# The Impact of Light Wavelength and Darkness on Metabolite Profiling of Korean Ginseng: Evaluating Its Anti-Cancer Potential against MCF-7 and BV-2 Cell Lines

**DOI:** 10.3390/ijms24097768

**Published:** 2023-04-24

**Authors:** Nooruddin Bin Sadiq, Hyukjoon Kwon, Nam Il Park, Muhammad Hamayun, Je-Hyeong Jung, Seung-Hoon Yang, Soo-Won Jang, Seda Nur Kabadayı, Ho-Youn Kim, Young-Joo Kim

**Affiliations:** 1Smart Farm Research Center, Korea Institute of Science and Technology (KIST), Gangneung 25451, Republic of Korea; nooruddin@kist.re.kr (N.B.S.); jhjung@kist.re.kr (J.-H.J.); t22607@kist.re.kr (S.N.K.); 2Department of Plant Science, Gangneung-Wonju National University, Gangneung 25457, Republic of Korea; nipark@gwnu.ac.kr; 3Center of Biomaterials, Korea Institute of Science and Technology (KIST), Gangneung 25451, Republic of Korea; 120053@kist.re.kr; 4Department of Botany, Abdul Wali Khan University Mardan, Mardan 23200, Pakistan; hamayun@awkum.edu.pk; 5Department of Medical Biotechnology, College of Life Science and Biotechnology, Dongguk University, Seoul 04620, Republic of Korea; shyang@dongguk.edu; 6Korean Ginseng Company (KGC), 71 Beotkkot-gil, Daedeok-gu, Daejeon 34337, Republic of Korea; swjang@kgc.co.kr; 7Division of Bio-Medical Science and Technology, KIST School, University of Science and Technology (UST), Daejeon 34113, Republic of Korea

**Keywords:** *Panax ginseng*, ginsenosides, metabolomics, hydroponics, anti-cancer potential, MCF-7, BV-2

## Abstract

Korean ginseng is a source of functional foods and medicines; however, its productivity is hindered by abiotic stress factors, such as light. This study investigated the impacts of darkness and different light wavelengths on the metabolomics and anti-cancer activity of ginseng extracts. Hydroponically-grown Korean ginseng was shifted to a light-emitting diodes (LEDs) chamber for blue-LED and darkness treatments, while white fluorescent (FL) light treatment was the control. MCF-7 breast cancer and lipopolysaccharide (LPS)-induced BV-2 microglial cells were used to determine chemo-preventive and neuroprotective potential. Overall, 53 significant primary metabolites were detected in the treated samples. The levels of ginsenosides Rb1, Rb2, Rc, Rd, and Re, as well as organic and amino acids, were significantly higher in the dark treatment, followed by blue-LED treatment and the FL control. The dark-treated ginseng extract significantly induced apoptotic signaling in MCF-7 cells and dose-dependently inhibited the NF-κB and MAP kinase pathways in LPS-induced BV-2 cells. Short-term dark treatment increased the content of Rd, Rc, Rb1, Rb2, and Re ginsenosides in ginseng extracts, which promoted apoptosis of MCF-7 cells and inhibition of the MAP kinase pathway in BV-2 microglial cells. These results indicate that the dark treatment might be effective in improving the pharmacological potential of ginseng.

## 1. Introduction

Ginseng is a medicinal herb used worldwide for its pharmacological and therapeutic potential [1]. The plant is commonly referred to as “Panax”, a term derived from the Greek word “Panacea”, meaning “cure-all” [2,3,4]. For centuries, Korean ginseng has been used in traditional medicine to treat various ailments, including fatigue [5], stress [6], anxiety [7], and inflammation [8]. It is believed to possess adaptogenic properties, helping the body adapt to stress and improving overall health. Additionally, Korean ginseng has gained attention in modern medicine for its potential pharmacological properties. It contains several active compounds, including ginsenosides, which exhibit anti-inflammatory [9], anti-cancer [10], and antioxidant effects. Research has also suggested that Korean ginseng may have potential benefits for cognitive functioning [11], cardiovascular health [12], and immune system functioning [13].

Ginseng can be successfully grown using hydroponics, a water-nutrient-based system used to grow plants in the absence of soil. Ginseng growth can be accelerated using nutrient-enriched hydroponic media. For example, four months of cultivation in a hydroponic system can easily produce roots equal in size to those of two-year-old ginseng plants grown using conventional methods [14].

Light is one of the major abiotic factors that influence the biosynthesis of secondary metabolites in plants [15]. Light-dependent processes, such as photosynthesis, can be modified under controlled lighting conditions, where the intensity, wavelength, and photoperiod are regulated [16]. Light-emitting diodes (LEDs) are mostly used as an artificial light source, one which improves the quality of food products in commercial agriculture [17] and upregulates the biosynthesis of defense-related secondary metabolites [18]. LEDs are efficient sources of light for growing plants and have many benefits, including improvement of photosynthesis in plants, specific bandwidth selection, long life, and reduced heat production [19]. White, blue, and red lights improve signal transduction and enhance the production of important secondary metabolites [20]. Specific wavelengths, such as red and blue, are good elicitors for the improved production of bioactive compounds, including phenols, carotenoids, and antioxidants [21]. Meanwhile, blue light is an effective pre-harvest treatment that delays fruit ripening and softening, thus improving the shelf life of different climacteric crops [22]. LED use is also recommended during storage or transportation to minimize post-harvest losses and maintain product quality [23].

In women, breast cancer is the most common type of malignant cancer found worldwide, with approximately 1.8–2 million new cases reported each year globally, and >60% of deaths occurring in underdeveloped countries [24,25]. Moreover, breast cancer is the most commonly-diagnosed cancer in developed countries, accounting for 30% of the total cancer diagnoses, and has an estimated death rate > 15% in women [26]. TNBC (triple negative breast cancer) is the most fatal form of breast cancer, as it has a high rate of proliferation and heterogeneity and no approved targeted molecular therapies are available against this cancer subtype [27]. In recent years, natural products have proven to be the most effective and safe compounds for the treatment of cancer. Natural products modulate the cancer microenvironment and intervene within cell signaling cascades [28]. Previous reports have suggested that ginsenosides, such as Rg3, Rh2, and Rh4, exhibit anti-tumor potential by inducing apoptosis in tumor cells and inhibiting cell proliferation and signaling pathways [29].

Microglial cells are macrophage-like cells that belong to the central nervous immune system. The main task of the microglial cells is to protect the brain by producing various neurotrophic factors to support neuronal cells during brain damage [30]. The overactivation of microglial cells in the central nervous system results in the overproduction of inflammatory mediators, such as pro-inflammatory cytokines and reactive oxygen species, which can be associated with the development of various neurodegenerative diseases, including Parkinson’s disease, multiple sclerosis, Huntington’s disease, and Alzheimer’s disease [31]. Several studies have reported that ginsenosides have neuroprotective effects against lipopolysaccharide (LPS)-induced microglial cells [32,33].

In ginseng plants, the quantity, quality, and total number of ginsenosides vary depending on both biotic and abiotic factors [34,35]. In addition, processed ginseng with higher content of Rg5, Rg3, and Rd ginsenosides exhibits higher anti-tumor potential than does unprocessed ginseng [36,37]. Therefore, this study aimed to determine the impact of different light treatments (i.e., blue-LED, dark, and white fluorescent (FL) light) on the anti-cancer potential of Korean ginseng extracts. MCF-7 (human breast cancer) and BV-2 (microglial cancer cells) cell lines were used to determine the effect of the treatments on the anti-cancer potential of ginseng plants cultivated in a hydroponic system. Ginsenosides and primary metabolites were profiled to reveal the complex interactions between primary metabolites, ginsenosides content, and therapeutic effects. The metabolic changes in ginseng plants growing under different light conditions might determine the specific characteristics of each light treatment.

## 2. Results

### 2.1. Analysis of Ginseng Metabolome by GC-TOF-MS

Korean ginseng is a source of proteins and terpenoids, particularly ginsenosides. A discriminatory holistic approach was applied in this study to compare the metabolite profiling of ginseng samples from the dark treatment with that of samples from the blue-LED treatment and FL control via GC-TOF-MS. Biological variance among treatments was assessed using three independent biological replicates performed under uniform conditions. Metabolite changes were observed among the hydroponically-grown ginseng plants from the different treatments (Table 1). A total of 101 metabolites were identified in the three treatments, including organic and amino acids, carbohydrates, and lactones (Table 1).

The dark-treatment samples were enriched in organic acids (36.62%), carbohydrates (33.96%), lactones (14.81%), and amino acids (13.47%). The blue-LED samples were rich in carbohydrates (46.16%), organic acids (24.57%), lactones (21.24%), and amino acids (6.10%). Meanwhile, the FL-control samples were enriched in carbohydrates (57.28%), organic acids (20.99%), lactones (15.61%), and amino acids (4.80%). Specific markers of the treatments were revealed after the multivariate analysis (Figure 1). Metabolite changes occurred during the biosynthesis of ginsenosides and their accumulation in hydroponically-grown ginseng under short-term exposure to darkness and various light wavelengths. The metabolite profiling of the dark-treated ginseng in this study was similar to that reported previously by Park et al. [38]. Carbohydrate content was previously reported as a specific marker of the white FL treatment, as it is higher in Korean ginseng than in American ginseng [39].

A principal component analysis (PCA) model was utilized to extract the most important variables or features from a large dataset of ginseng extract compounds obtained through GC-TOF-MS analysis. By reducing the dimensionality of the data and identifying the most significant compounds that had contributed to the variances in the dataset, the PCA model can aid in interpreting and visualizing complex data structures. Figure 1A shows the score plot of PCA derived from the GC–MS profiling of the treated ginseng plants. The first and second components of the PCA score plot explained 70.5% and 18.9% of the variation in the metabolite content among the different ginseng samples, respectively (Figure 1A). The dark-treated ginseng samples were fully distinguished in the PCA plot compared to the blue-LED and FL-control samples. The blue-LED samples were close to those of the FL control, suggesting that the blue-LED treatment and FL control shared considerable similarities when compared to the dark treatment.

The PCA scatter plot was used to identify the metabolites that segregated the dark and blue-LED treatments from the FL control (Figure 1B). The dark treatment could be differentiated from the FL control by its high levels of propanedioic acid, propanoic acid, L-glutamine, ferulic acid, L-aspartic acid, L-lysine, talonic acid, L-alanine, methionine, D-mannose, and ethyl methyl malonate. Moreover, the levels of serine, fructose, and tartaric, mannoic, pentanoic, and ribonic acids were higher in the blue-LED treatment than in the FL control. Correlation analysis was performed to understand the relevance of the metabolite compositions of the different treatments (Figure 1C). A heat map was used for the quantitative analysis of metabolites by using a z-score, and metabolites were plotted on a red-blue color scale (Figure 1D). The PCA and heat map patterns of the dark-treated samples were significantly different from that of the blue-LED and FL-control samples.

The quantitative differences in metabolites among treatments are shown in Figure 2. The content of amino acids, such as L-glutamine, asparagine, L-aspartic acid, and L-tryptophan, was significantly higher in the dark treatment than in the FL control. Amino acids (Figure 2A) play an important role in the conversion of major ginsenosides to minor ginsenosides [40]. Similarly, the content level of sugars (Figure 2B), including D-mannose, D-erythrotetrofuranose, and L-sorbopyranose, was significantly higher in the dark treatment than in the FL control. Among the organic acids (Figure 2C), glyceric, malic, and citric acids were higher in the dark treatment than in the FL control, which is consistent with the findings of Seong et al. [38].

### 2.2. Ginsenosides Composition under Different Treatments

We determined the ginsenosides content of the ginseng extracts used in each treatment. Total ginsenosides, including Rg1, Re, Rb1, Rc, Rb2, Rd, Rg6, F4, and F2, were quantified using HPLC-UV-VIS. Each compound was identified by comparing its retention time with that obtained from the chromatograms of the mixed ginsenoside standards. FL-control samples contained Rb1, Rb2, Rc, Rd, Re, Rg1, Rg6, F2, and F4 as the major ginsenosides (Figure 3A). After seven days of dark treatment, the contents of Rg1, Re, Rb1, Rc, Rb2, and Rd remarkably increased when compared with those of the blue-LED treatment and FL control. Upregulation of β-glucosidase enzymatic activities has been effective in increasing the Rb1, Rb2, Rc, and Rd content in ginseng [41]. The content of Rd ginsenosides has been found to be higher in red ginseng when compared to FL [42]. In this study, F2 levels were higher in the FL control than in the blue-LED treatment (Figure 3B). Moreover, Re and F2 were the most abundant ginsenosides among all treatments. The abundance of Re in ginseng was also reported by Xu et al. [43]. Therefore, the exposure to short-term dark treatment improved the overall ginsenoside content in ginseng plants.

### 2.3. Dark-Treated Ginseng Extract Reduces the Viability of MCF-7 Cancer Cells

The anti-cancer activity of the various ginseng extracts was tested on the MCF-7 cell line. The MCF-7 cell line contains progesterone, estrogen, and glucocorticoid receptors [44]. Due to its ability to retain the characteristics of the mammary epithelium, it is mostly used as a model system for the apoptosis of cancer cells [45,46]. The dark-treated ginseng extract had a strong potential to induce apoptosis in the MCF-7 cancer cells among all treatments. To select a suitable nontoxic range of drug concentrations in cell culture, the ginseng extracts from all treatments were assessed for their effects on the viability of MCF-7 cells by MTT reduction assay for 24 h (Figure 4A). The cytotoxic effects of ginseng extracts were first measured at a concentration of 100 μg/mL (Figure 4A). The dark-treated ginseng extract showed significant cytotoxicity (IC_50_ = 87.1 μg/mL) in the MCF-7 cell line. In general, an extract concentration that reduces cell viability to <50% is considered to be an effective concentration, and this was only achieved by the dark-treated ginseng extract (Figure 4B). To examine whether the anti-proliferative effect of the dark-treated ginseng extract was involved in apoptotic cell death, we examined PARP protein status in MCF-7 cells exposed to the extracts. Apoptosis is a crucial process in various biological events, including development, immune response, and maintenance of cellular balance. This process is characterized by significant alterations in cellular morphology, such as chromatin condensation, membrane blebbing, nuclear breakdown, and the emergence of membrane-bound apoptotic bodies. Additionally, it involves inter-nucleosomal DNA fragmentation and cleavage of poly(ADP-ribose) polymerase (PARP) [47]. PARP is an enzyme that catalyzes poly(ADP-ribose) conversion of several nuclear proteins, using NAD as a substrate [48]. It was suggested that PARP contributes to cell death by depleting the cell of NAD and ATP, since it is activated by binding to DNA ends or strand breaks [49]. Furthermore, PARP is cleaved into two fragments of 89 and 24-kDa during drug-induced apoptosis in various cells. These fragments contain the active site and DNA-binding domain of the enzyme, respectively, and essentially inactivate the enzyme by destroying its ability to respond to DNA strand breaks [50].

As shown in Figure 4C, the dark-treated ginseng extracts induced PARP cleavage in a dose-dependent manner. These findings suggested that the dark-treated ginseng extract inhibited cell viability by inducing apoptotic signaling in breast cancer cells. Western blotting clearly showed the ability of the dark-treated ginseng extract to induce PARP cleavage, as indicated by the diminution of its parent band (116 kDa) and the concomitant accumulation of its cleavage fragment (89 kDa) (Figure 4C). The interrelationship among caspase-3 activation, PARP degradation, DNA fragmentation, and apoptosis induction has been well defined in the literature [51,52,53]. PARP degradation and apoptosis induction in MCF-7 cells provided additional insights into the mechanisms of action of the dark-treated ginseng extract. GAPDH was used as an internal control.

### 2.4. Dark-Treated Ginseng Extract Inhibits the Pathway of NF-κB Signaling

We examined a key transcription factor that modulates the expression of a wide range of pro-inflammatory genes to investigate the impact of the treatments on the effectiveness of ginseng extracts against the activation of the NF–κB signaling pathway. As shown in Figure 5, the ph-p65 levels were highly induced upon LPS administration; however, the level of p65 phosphorylation was significantly inhibited by the dark-treated ginseng extract in a dose-dependent manner (1–25 µg/mL). The dark treatment displayed a stronger inhibitory activity on p-65 protein expression than did the blue-LED treatment or the FL control. Next, the effects of the treatments on the IκB–NF–κB signaling pathway were measured using western blot analysis. Cells exposed to the dark-treated ginseng extract showed increased expression levels of IκB-α and reduced phosphorylation levels of IκB-α compared to LPS-stimulated microglia, suggesting that the dark treatment suppressed the LPS-induced IκB-α phosphorylation and degradation (Figure 5). Our results showed similar suppressive effects on cancer cells than those reported in the literature for other treatments [54]. The dark-treated ginseng extracts decreased LPS-induced NF-κB p65 nuclear translocation at concentrations ranging from 1–25 μg/mL.

### 2.5. Activation of MAPK Signaling Was Inhibited by Dark Treated Ginseng Extract

A crucial mediator in the activation of the microglia’s secreted pro-inflammatory cytokine production is the MAPK signaling pathway. We examined the effects of the dark-treated ginseng extracts on phosphorylated MAPK in LPS-treated microglia in order to assess their impact on the activation of inflammatory mediators through MAPK. Stimulation of BV-2 microglia with LPS increased the phosphorylation levels of ERK, JNK, and p38 MAPKs. Dark-treated ginseng extracts significantly attenuated the LPS-induced phosphorylation of representative MAPKs, including ERK, JNK, and p38, in a dose-dependent manner (Figure 5A,B). Activation of MAPK signaling was also inhibited by the dark-treated ginseng extracts (Appendix A)

## 3. Discussion

The advancement of injury, infection, toxicity, or autoimmunity in the central nervous system (CNS) is heavily influenced by neuroinflammation [55]. This occurs when microglia become activated and release neurotoxic molecules and pro-inflammatory cytokines, causing progressive neuronal cell degeneration [56]. Therefore, reducing the excessive activation of microglia is a promising approach for the alleviation of the inflammatory response in the nervous system [57]. The use of natural compounds as a potential source for the development of new medicines to alleviate neuroinflammation and treat neurological disorders has gained significant recognition.

The anti-inflammatory properties of ginseng extracts have been established through the use of purified ginsenosides, including ginsenosides Rb1, Rg1, Rg3, Rh2, and compound K. These ginsenosides have been found to negatively regulate the expression of pro-inflammatory cytokines such as TNF-α, IL-1β, and IL-6, as well as enzymes including iNOS and COX-2, thus demonstrating their anti-inflammatory mechanism in M1-polarized macrophages and microglia [9]. Ginsenoside Rg3, one of the most important minor ginsenosides found in Korean ginseng, can significantly reduce the inflammatory response induced by LPS, as well as mediate TNF-α, and IL-6 in BV2 and primary microglial cells [30]. Ginsenoside Rh2 (GRh2) is a well-known constituent found in red ginseng, a plant which has been widely used in Korea and China for various medical purposes and as a dietary supplement. GRh2 has been reported to exhibit several beneficial biological effects, such as anti-inflammatory, antioxidant, and neuroprotective properties, among others. Its potential therapeutic applications have been studied extensively in recent years [58]. Rh3 is a bacterial byproduct of heat-processed ginseng’s main component, Rg5. Rh3 has anti-inflammatory effects by means of inhibiting the expression of iNOS and pro-inflammatory cytokines such as TNF-α and IL-6 and enhancing the expression of anti-inflammatory hemeoxygenase-1. Rh3 also inhibits NF-κB through SIRT1 upregulation and enhances Nrf2 DNA-binding activities. It activates AMPK phosphorylation and inhibits Akt and JAK1/STAT1 pathways induced by LPS [59].

In our study, we investigated the impact of darkness and blue LED treated ginseng extract on five different cell lines (Appendix A) and found that dark-treated hydroponic ginseng could inhibit cytokine expression in LPS-stimulated microglia to exert neuroprotective activity. Our results suggest that different experimental environments and conditions may affect pharmacological potential; thus, we will study the impact of different LED treatments in future studies. LPS is a widely used method for studying neuroinflammation in laboratory experiments, both in vitro and in vivo. The LPS model is advantageous due to its simplicity and consistent activation of inflammation. Following LPS administration, an excess of pro-inflammatory cytokines can soon be detected [60]. In our research, we found that a short period of dark treatment in hydroponic systems resulted in the high level of anti-inflammatory activity in BV-2 microglia cells exposed to LPS through various signaling pathways. Our findings indicate that an ethanolic extract obtained from ginseng subjected to dark treatment was effective in inhibiting the phosphorylation of NF-κB P65 and the expression of NF-κB P50 in the nucleus of microglia cells induced by LPS. Additionally, the phosphorylation of MAPKs was suppressed in a concentration-dependent manner.

In conclusion, the present study revealed that LED treatment can improve the overall pharmacological potential of Korean ginseng, with dark treatment being the most effective, and that these treatments could inhibit the expression of cytokines in LPS-induced microglia cells in in-vitro conditions. Further research found that dark-treated ginseng extract expressed higher anti-inflammation activity by inhibiting the NF–κB and MAPK signaling pathways. Ginsenosides content plays an important role in the improved neuroprotective potential of dark-treated ginseng, as the content of ginsenosides Rd, Rb2, and Rc were significantly improved. Ginsenoside Rd is further converted to ginsenosides Rg3 and F2, which already reported having higher neuroprotective potential [61]. Moreover, hydroponically-grown ginseng within improved environmental conditions can reduce the neuron damage induced by neuroinflammation and can be the best natural remedy for curing cancers such as breast cancer.

## 4. Materials and Methods

### 4.1. Plant Material

One-year-old ginseng (*Panax ginseng* cv. Geumpung) root seedlings were obtained from the Korean Ginseng Company and grown in a hydroponic system. The roots were first soaked in water and incubated for 48 h in moist conditions to germinate the bud. Then, they were transplanted to a nutrient bath under white light in a vertical farming facility. Light intensity was set at 65–70 µmol/m^2^/s with a photoperiod of 16 h/8 h (day/night) at 24 °C and 18 °C, respectively. Polyurethane sponge was used as a supporting medium for the deep-water culture-type hydroponic system. RDA (Rural Development Administration) approved nutrient solution for ginseng was supplied to the seedlings. The nutrient solution was composed of NO_3_-N 12.5, NH_4_-N 1.3, K 8.5, Ca 4.5, Mg 2.3, PO_4_-P 3.5, SO_4_-S 2.0, Fe-EDTA 0.8, Mn 0.6, B 0.6, Cu 0.02, Mo 0.06, and Zn 0.06 mg/L. The electrical conductivity of the media was maintained at 2.5 mS/cm with pH 6.8. After 4 weeks of growth, the ginseng seedlings were shifted to a LED chamber to conduct light treatments [62].

### 4.2. Light Treatments and Ginsenosides Extraction

Ginseng plants underwent three different treatments: dark, blue-LED (465–470 nm), and white FL light. Light treatments lasted for 10 days, with the application of a light intensity of 65–70 µmol on a 24/7 schedule, whereas the dark treatment lasted for 7 days. The whole ginseng plants, including both aerial parts and roots, were harvested after each treatment and immediately stored at −80 °C, followed by freeze-drying for 48 h in a freeze-dryer (FDCF-12006 Operon Co., Ltd., Gimpo, Republic of Korea). Freeze-dried plant powder was extracted thrice with 50 mL of extraction solvent (70% MeOH) by sonication (UCP02, Jeiotech, Daejeon, Republic of Korea) for 60 min [63]. After filtration using filter paper (0.2 μm, ADVANTEC, Dublin, CA, USA), evaporation method was used to remove solvent, and the dried residue was dissolved in 100% dimethyl sulfoxide (DMSO) at 40 mg/mL concentration. Each treatment had three biological replicates.

### 4.3. Ginsenosides Analysis by HPLC—Evaporative Light Scattering Detection (ELSD)

To determine the ginsenoside content, the ginseng extracts were diluted to 1 mg/mL using 100% MeOH [64,65,66]. Using a SIL-9A auto-injector, a volume of 10 L was injected into the HPLC system (Hitachi L-6200 pump, Tokyo, Japan) connected to a Sedex 75 ELSD (Sedere, Vitry-sur-Seine, France) (Shimadzu, Japan). All separations were performed using an Agilent Technologies (Palo Alto, CA, USA) Zorbax SB-Aq C18 column (4.6 mm 150 mm, 5 m particle size). The high-pressure liquid chromatographic (HPLC) conditions were as follows: solvent A, water; solvent B, acetonitrile. B 20–22% (0–5 min); 22–25% (5–7 min); 26–30% (27–30 min); 30–35% (30–40 min); 50–70% (45–60 min); 70–85% (60–61 min); 85–100% (61–90 min). Nebulizer for nitrogen gas was adjusted to 2.5 bar, and ELSD was set to a probe temperature of 75 °C. A total of 5 μg of each ginsenoside standard was injected for HPLC analysis [66].

### 4.4. Sample Preparation for Primary Metabolites Profiling

Freeze-dried sample powders (10 mg) were dissolved in a GC–MS extraction solvent containing water, methanol, and chloroform in ration of 2.5:1:1, V/V/V, ribitol and 5α-cholestane as internal standards. The samples were extracted at 37 °C, 1200 rpm for 30 min, followed by centrifugation at 13,000 rpm for 3 min. Top part, (polar phase 8 mL) was transferred to a new tube, and 4 mL double distilled water was added, centrifuged for 3 min at 13,000 rpm and were dried using a freezing dryer. Derivatization process was performed based on standard procedure using trimethylsilyl and methoxyamine hydrochloride [67,68].

### 4.5. Profiling of Primary Metabolites

Gas chromatography (GC) in conjunction with a time-of-flight mass spectrometer (TOF-MS) were used to investigate primary metabolites (LECO Pegasus GC HRT, Leco Corp., St. Joseph, MI, USA). At 230 °C, 1 µL by volume sample was injected into the machine with the mode being split with 1:25 ratio. Helium, the carrier gas, with 1 mL/min flow rate RTX-5MS (30 m 0.25 mm; 0.25 m film thickness) was the column used. The oven temp was first set at 80 °C for 2 min before rising at a rate of 15 °C/min to 320 °C for 10 min. The ion source and transfer line were both adjusted at 200 °C and 250 °C, respectively. A total of 25 spectra were acquired every second at a rate of TOF-MS acquisition that was delayed for 200 s. The electron energy and detector voltage were set at 70 eV and 1600 V, respectively [69].

### 4.6. Cell Cultures

In accordance with the five most common cancer varieties in Korea, we used five established human cancer cell lines. The American Type Culture Collection (ATCC, Manassas, VA, USA) provided the human lung carcinoma cell line A549, human breast cancer cell line MCF-7, human colorectal carcinoma cell line HCT116, and human gastric cancer cell line AGS (ATCC, Manassas, VA, USA). The Korean Cell Line Bank sold the human hepatocellular carcinoma cell line Huh7 (Seoul, Korea). RPMI 1640 (Gibco, Waltham, MA, USA) media was used for regular subcultures of cell with 15% bovine fetal serum, 150 U/mL of penicillin, and 100 nanograms per milliliter of streptomycin in a humid environment with 37 °C and 5% carbon dioxide [70].

### 4.7. Cell Viability Assay

According to the manufacturer’s instructions, the EZ-Cytox test kit (Daeil Lab Service, Seoul, Korea) was used to assess the cytotoxic effects of the ginseng extracts on the examined cell lines. In a 96-well plate, triplicates of exponentially developing cells were planted at a density of 1.0 104 cells per well. The cells were given 1, 5, and 25 g/mL of the extracts the following day. The cells were cultured for 1 h with 10 L of the kit reagent after 24 h of incubation. A microplate reader operating at a 450 nm wavelength was used to scan cell cultures to determine cell viability. A quantum of 0.5% (*v*/*v*) DMSO-containing culture solutions were used to expose control cells to. With extract concentration on the *x*-axis and the relative cell viability on the *y*-axis of a non-linear regression curve, the concentration needed to inhibit cell viability by 50% (IC50) was graphically estimated [71].

### 4.8. Preparation of Nuclear Extract, Whole-Cell Lysates and Immunoblot Analysis

Ginseng extracts from the various light treatments were given to MCF-7 cells that had been cultured in 6-well plates for 24 h. Then, using RIPA buffer (Cell Signaling Technology, CST, Danvers, MA, USA) together with a 1 protease inhibitor cocktail (Roche, Mannheim, Germany) and 1 mM phenylmethylsulfonyl fluoride, whole-cell lysates were created in accordance with the manufacturer’s instructions. Proteins (whole-cell extracts, 30 g/lane) were electrophoretically separated from one another on NuPAGE 4–12% Bis-Tris gels (Invitrogen, Waltham, MA, USA), blotted onto PVDF transfer membranes, and examined with primary and secondary antibodies that are specific for the epitopes in question. Western blotting detection reagents from Thermo Fisher Scientific, Waltham, Massachusetts, USA, and a LAS 4000 imaging system were used to detect bound antibodies (Fujifilm, Tokyo, Japan). Rabbit anti-PARP (9542; CST) and rabbit anti-GAPDH were the principal antibodies employed (2118; CST). The HRP-linked secondary antibody was goat anti-rabbit IgG (7074, CST) [72].

Before starting nuclear and cytoplasmic extraction, two lysate buffers were prepared namely buffer A (10 mM HEPES, 10 mM KCl, 1 mM EDTA, 0.2 mM EGTA, 1 mM PMSF, 1 mM DDT, 0.5% NP40, and a protease inhibitor cocktail [Sigma-Aldrich, St. Louis, MO, USA] in HPLC grade water) and buffer B (0.1 mM EDTA, 0.1 mM EGTA, 20 mM HEPES, 420 mM NaCl, 1 mM PMSF, 1 mM DDT, and protease inhibitor cocktail in RIPA buffer [Sigma-Aldrich]) [73]. For cytoplasmic extraction cells were treated with varying concentrations of isodojaponin D (1, 5, or 10 μg/mL) in the presence or absence of LPS (1 μg/mL) for 24 h. After LPS stimulation, buffer A was used for cell lyses. Homogenate were vortexed for 1 min followed by centrifugation for 5 min at 20,000× *g*, supernatant was collected as cytoplasmic extract. Resulting residual pellets were used for nuclear extraction using 50 μL of buffer B. After homogenization and centrifugation (20,000× *g* for 5 min) supernatant was collected as nuclear extract. Both extracts were stored at −70 °C [74].

### 4.9. MTT Assay

The MTT assay was used to determine cell cytotoxicity. A quantum of 0.2 × 10^4^ cells/well of cells were sown in 96-well culture plates, and the cells were left to adhere for 24 h. Different concentrations of ginseng extracts, such as dark, blue-LED, and FL (e.g., 5 mg/mL, 10 mg/mL, 15 mg/mL, and 20 mg/mL), were applied to each well of culture media. A total of 30 µL of MTT (3-(4,5-dimethylthiazol-2-yl)-2,5-diphenyl tetrazolium bromide) stock solution (3 mg/mL) were applied to each well at the appropriate time intervals. DMSO was applied to the cells after two hours of culture at 37 °C to dissolve the formazan crystals. Using a microplate reader (EL800, Biotek Instruments Inc., Winooski, VT, USA), the absorbance of the cell culture was measured at a 540 nm wavelength [75].

### 4.10. Western Blot Analysis

Bradford method (Bio-Rad, Hercules, CA, USA) was used to assay total proteins in cell extracts or nuclear and cytoplasmic extracts. For SDS-PAGE, 30 μg of protein from cell lysates was separated according to size. Equal amounts of protein for each sample were heated with 2X Laemmli sample buffer and loaded onto 12% or 7.5% polyacrylamide gels. Proteins were then transferred onto a PVDF (Millipore, Billerica, MA, USA) membrane for 2.4 h at 400 mA. The membrane was blocked with 5% nonfat dry milk and incubated with primary antibody overnight at 4 °C. The antibodies used were NF-κB p50 (1:1000, Santa Cruz, Santa Cruz, CA, USA), NF-κB p65 (1:1000, Santa Cruz), c-Jun Nterminal kinase (JNK, 1:1000, Cell Signaling), p-JNK (1:1000, Cell Signaling), p38 (1:1000, Santa Cruz), Phospho-p38 MAPK (1:1000, Cell Signaling), extracellular-signal-regulated kinase (ERK, 1:5000, Cell Signaling), pERK (1:1000, Cell Signaling), and β-actin (1:100,000, Sigma Aldrich). After incubation, the membranes were washed with TBS-T buffer and incubated again with the respective secondary antibodies conjugated with peroxidase (Sigma-Aldrich). Secondary antibodies conjugated to horseradish peroxidase (1:2500, Santa Cruz), followed by enhanced chemiluminescence (ECL Western Blotting Systems, GE Healthcare, Little Chalfont, BKM, UK) reagents (Amersham Biosciences, Piscataway, NJ, USA), were used for the detection of proteins. The western blot scanning results showed a linear curve in the range used for each antibody. The bands observed on the membrane were representative of the groups. To quantify the proteins, normalization with a control group was performed [76].

### 4.11. Statistical Analysis

Data were expressed as mean ± standard error (SEM) of the mean of n observations. Principal component analysis (PCA) was performed using MetaboAnalyst version 5.0 to identify the different metabolites produced in each treatment. Orthogonal partial least squares discriminant analysis (OPLS-DA) was performed using SIMCA-P, version 15.0.2 (Umetrics, Umea, Sweden). All experiments were repeated at least thrice, and all data were compiled from a minimum of three replicates. Significant differences between treated and untreated cell lines were analyzed by post-hoc Tukey’s test followed by two-way ANOVA. *p*-values < 0.05 were considered significantly different (for three biological replicates) between each compared pair groups.

## 5. Conclusions

A GC–MS-based metabolomics approach revealed trends in the accumulation of various metabolites in ginseng in response to darkness and different light treatments. The short-term dark treatment increased the organic acid content in ginseng and decreased carbohydrate content by converting monosaccharides into complex secondary metabolites, such as ginsenosides Rd and Rb. The abundant Rb1 and Rb2 in the dark treatment served as major substrates for the cell enzymatic conversion to Rg3, which, together with other minor ginsenosides, increased the chemo-preventive and neuroprotective potential of the dark-treated ginseng. Our investigation revealed that the dark-treated ginseng extract induced apoptosis in MCF-7 cells. Western blot analysis was used to evaluate the expression and cleavage of PARP protein as well as caspase 3 expression, which was regulated in a dose-dependent manner. Dark-treated ginseng extracts prevented LPS-induced IκB-α/β phosphorylation, which, in turn, inhibited the translocation of NF-κB-dependent pathways. This finding suggests that the dark treatment is effective in improving the overall metabolite profiling of ginseng. Moreover, the short-term dark treatment is highly effective in enhancing the biological conversion of major ginsenosides to minor ginsenosides, which in turn enhances the pharmacological potential of ginseng. The GC-MS-based metabolomics approach is an efficient tool for metabolite profiling in plants. Metabolomics-based profiling tools may be useful for improving the quality of ginseng plants. Further in-depth investigations are recommended to identify specific genes, whose upregulation could increase the content of minor ginsenosides in dark-treated ginseng.

## Figures and Tables

**Figure 1 ijms-24-07768-f001:**
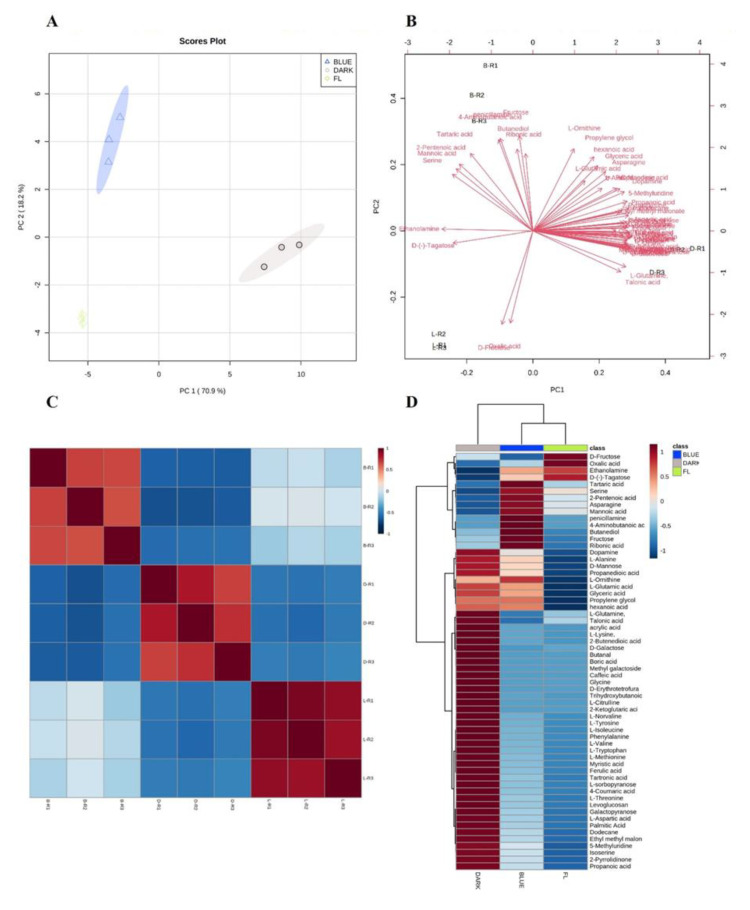
(**A**) Principal component analysis score; (**B**) loading plot; (**C**) correlation analysis; and (**D**) heat map from 53 significantly different detected compounds, as compared to control. Blue, gray, and green dots represent blue, dark, and FL treatment in the PCA plot, respectively. The heat map in D panel was presented with a red—blue color scale with red representing a higher metabolites level and blue for lower metabolites levels. A value of *p* < 0.05 was used to determine statistically significant difference.

**Figure 2 ijms-24-07768-f002:**
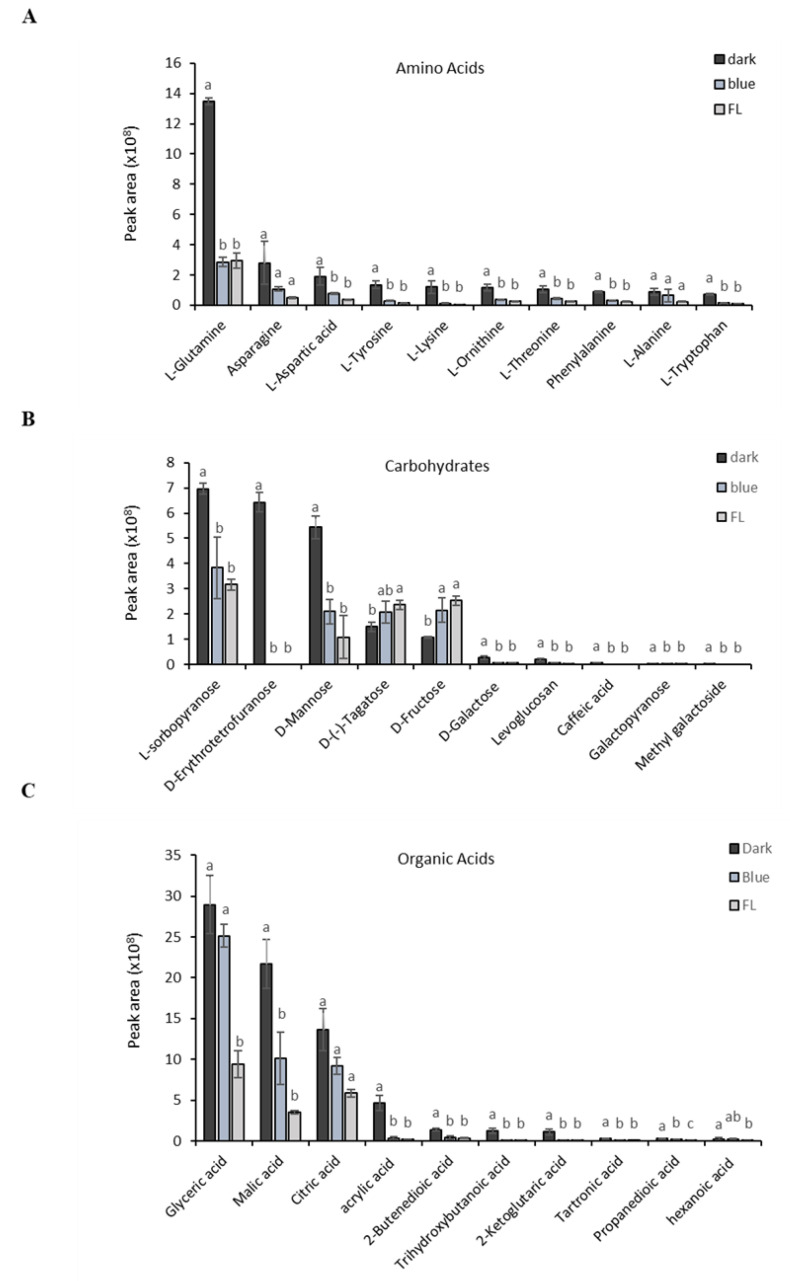
Primary metabolites contents in dark treatment: (**A**) amino acids, (**B**) carbohydrates and (**C**) organic acids. Values are means± standard deviation. Values with the same superscript letter are not significantly different at *p* < 0.05 level.

**Figure 3 ijms-24-07768-f003:**
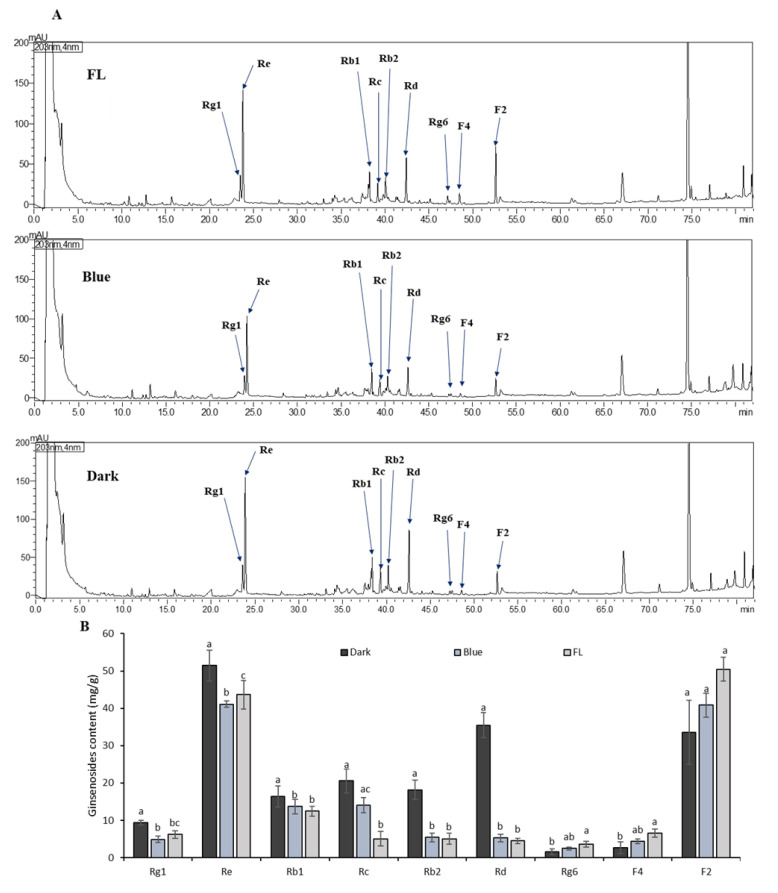
HPLC chromatograms compression showing: (**A**) TIC of total ginsenosides in FL, blue—LED, and dark treatment; and (**B**) ginsenosides content in dark, FL and blue-LED treated ginseng. Column and error bars represent the means and standard deviation (n = 5). Columns with different letters for each specific ginsenosides are significantly different (*p* < 0.05).

**Figure 4 ijms-24-07768-f004:**
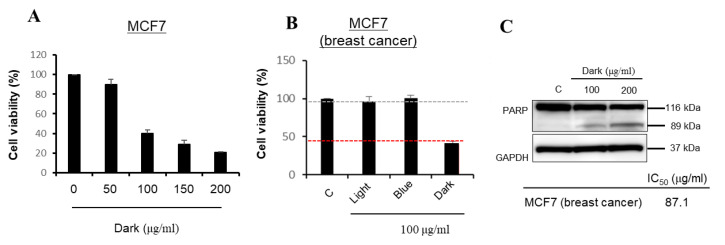
Anticancer effect of LED-treated ginseng on MCF7 cell: (**A**) cell viability assay for dark treatment; (**B**) comparison of cell viability among three different LED treatments; and (**C**) immunoblot analysis revealed the protein levels of PARP. GAPDH was used as an internal control. IC_50_ value of extract on MCF7 cell line. Each value represents the mean ± SD of three independent experiments.

**Figure 5 ijms-24-07768-f005:**
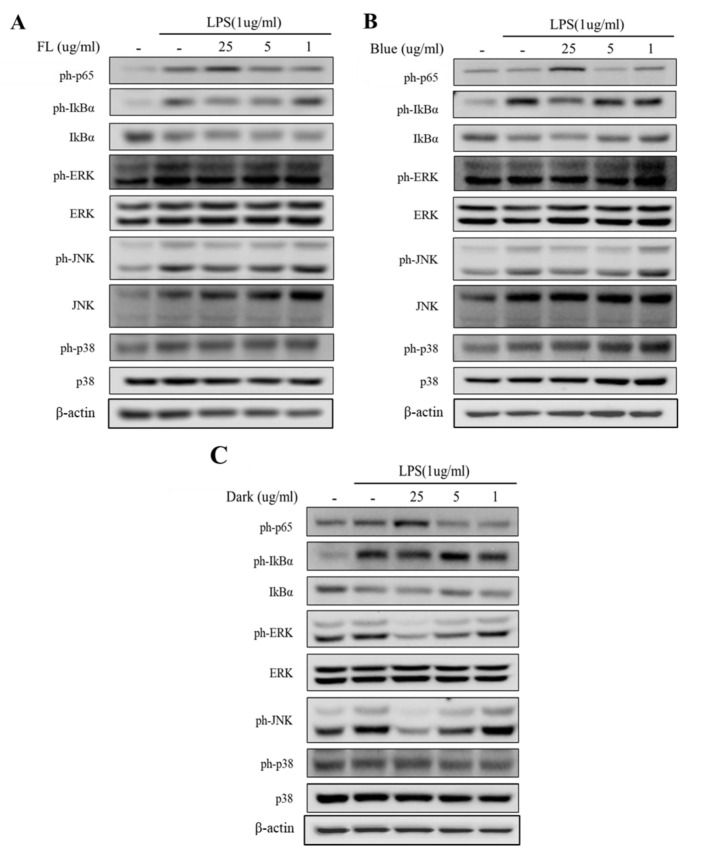
The inhibitory effect of LED-treated ginseng on MAPKs in LPS-stimulated BV2 cells: (**A**) FL as control, (**B**) blue-LED, (**C**) dark treatment. Western blot analysis on p38, ERK1/2 and JNK phosphorylation level in BV-2 cells. Cells were pre-treated with dark-, blue-, and FL-treated ginseng extract (1–25 µg/mL) for 2 h prior to LPS (1 µg/mL) exposure for 24 h. β-actin used as housekeeping protein.

**Table 1 ijms-24-07768-t001:** Significantly different primary metabolites as compared to control in dark and blue-LED treated samples by GC-TOF-MS (*p* < 0.05). Three biological replications were analyzed. ND means “not detected”.

NO	RT(SEC)	Identified Compound	Formula	Treatment	Peak Area	*p*-Value
Organic Acids						
1	274.6	Boric acid	H_3_BO_3_	FL	ND	
				Dark	55,158 ± 4979	0.003
				Blue	ND	
2	326.8	Ethyl methyl malonate	C_6_H_10_O_4_	FL	ND	
				Dark	2,648,561 ± 77,905	0.000
				Blue	907,763 ± 450,128	
3	414.5	Propanedioic acid	C_3_H_4_O_4_	FL	11,837,242 ± 963,277	
				Dark	26,345,064 ± 807,597	0.000
				Blue	20,531,543 ± 1,296,059	0.01
4	471	4-Aminobutanoic acid	C_4_H_9_NO_2_	FL	ND	
				Dark	ND	
				Blue	35,057,390 ± 5,863,536	0.001
5	490	Glyceric acid	C_3_H_6_O_4_	FL	941,334,205 ± 162,244,741	
				Dark	2,894,792,786 ± 755,578,224	
				Blue	2,514,331,459 ± 142,931,824	0.01
6	495.1	2-Butenedioic acid	C_6_H_8_O_4_	FL	3,388,837 ± 3,149,704	
				Dark	138,224,506 ± 14,064,424	0.008
				Blue	38,551,250 ± 20,640,438	
7	603	Trihydroxybutanoic acid	C_4_H_8_O_5_	FL	6,946,451 ± 1,197,940	
				Dark	102,284,588.33 ± 3,523,581	0.000
				Blue	9,506,910 ± 901,734.6	
8	628	Oxalic acid	C_2_H_2_O_4_	FL	15,000,978 ± 2,279,165	
				Dark	ND	
				Blue	5,048,996 ± 997,737	0.03
9	666.3	Hexanoic acid	C_6_H_12_O_2_	FL	6,361,231 ± 2,906,165	
				Dark	25,147,663 ± 3,163,963	0.03
				Blue	23,949,708 ± 4,094,744	0.003
10	703.3	Ribonic acid	C_5_H_9_O_6_	FL	10,727,733 ± 6,149,856	
				Dark	13,826,780 ± 866,318	
				Blue	23,317,685 ± 7,507,867	0.01
11	745.7	2-Ketoglutaric acid	C_5_H_6_O_5_	FL	5,572,062 ± 1,486,034	
				Dark	110,162,880 ± 33,201,327	0.04
				Blue	6,303,697 ± 4,026,419	
12	775.5	Acrylic acid	C_3_H_4_O_2_	FL	17,604,061 ± 1,032,006	
				Dark	463,504,027 ± 92,918,712	0.01
				Blue	3,5014,339 ± 14,354,648	
13	780.4	4-Coumaric acid	C_9_H_8_O_3_	FL	ND	
				Dark	8,441,648 ± 4,174,601	
				Blue	1,794,541 ± 258,199.4	0.008
14	785.4	Tartronic acid	C_3_H_4_O_5_	FL	2,751,360 ± 304,388	
				Dark	27,093,984 ± 4,230,208	0.001
				Blue	7,538,675 ± 1,832,029	0.05
15	826	Tartaric acid	C_4_H_6_O_6_	FL	3,5980,414 ± 2,407,682	
				Dark	ND	
				Blue	109,798,116 ± 3,291,842	0.000
16	854.5	Ferulic acid	C_10_H_10_O_4_	FL	ND	
				Dark	19,429,576 ± 7,087,994	0.05
				Blue	3,751,615 ± 607,332	0.01
17	871	2-Pentenoic acid	C_5_H_8_O_2_	FL	2,427,381 ± 154,542	
				Dark	ND	
				Blue	5,276,122 ± 243,489	0.005
18	936.2	Propanoic acid	C₃H₆O₂	FL	ND	
				Dark	4,142,119 ± 68,430	0.000
				Blue	1,682,445 ± 172,712	0.004
19	1484	Mannoic acid	C_6_H_12_O_7_	FL	103,339,880 ± 7,836,722	
				Dark	ND	
				Blue	204,832,544 ± 15,541,722	0.01
Amino Acids						
20	341.2	L-Valine	C_5_H_11_NO_2_	FL	648,024 ± 38,340	
				Dark	21,271,780 ± 5,902,237	0.03
				Blue	3,254,685 ± 1,830,534	
21	350.5	L-Alanine	C_3_H_7_NO_2_	FL	22,921,160 ± 2,208,680	
				Dark	98,293,721 ± 15,568,039	0.01
				Blue	64,064,382 ± 40,670,416	
22	506.2	Serine	C_3_H_7_NO_3_	FL	38,823,176 ± 3,976,486	
				Dark	ND	
				Blue	66,865,059 ± 8,216,446	0.01
23	512.4	Isoserine	C_3_H_7_NO_3_	FL	1,655,877 ± 625,229	
				Dark	5,951,890 ± 19,049	0.01
				Blue	3,295,161 ± 1,544,319	
24	521.4	L-Threonine	C_4_H_9_NO_3_	FL	26,596,544 ± 3,132,799	
				Dark	106,533,538 ± 20,954,160	0.03
				Blue	4,396,874 ± 792,504	0.02
25	562.2	L-Citrulline	C_6_H_13_N_3_O_3_	FL	ND	
				Dark	4,730,208 ± 335,083	0.002
				Blue	68,736 ± 53,704	
26	589.9	L-Aspartic acid	C_4_H_7_NO_4_	FL	36,974,682 ± 3,990,717	
				Dark	189,220,043 ± 58,130,937	0.01
				Blue	76,458,966 ± 3,877,746	0.006
27	591.1	L-Methionine	C_4_H_7_NO_4_	FL	3,918,664 ± 755,737	
				Dark	25,062,068 ± 579,209	0.000
				Blue	6,590,229 ± 635,920	
28	634.6	L-Ornithine	C_5_H_12_N_2_O_2_	FL	25,776,495 ± 2,926,910	
				Dark	114,425,820 ± 21,676,407	0.02
				Blue	36,284,134 ± 3,763,759	
29	636.9	L-Glutamic acid	C_5_H_9_NO_4_	FL	ND	
				Dark	30,258,688 ± 4,385,870	0.008
				Blue	22,155,999 ± 26,019,046	
30	589.9	L-Aspartic acid	C_4_H_7_NO_4_	FL	36,974,682 ± 3,990,717	
				Dark	155,886,710 ± 18,716,099	0.01
				Blue	76,458,966 ± 3,877,746	0.006
31	644.7	Phenylalanine	C_9_H_11_NO_2_	FL	22,839,808 ± 2,262,238	
				Dark	89,290,303 ± 6,606,822	0.005
				Blue	30,596,843 ± 2,963,083	0.04
32	676.7	L-Lysine	C_6_H_14_N_2_O_2_	FL	6,874,649 ± 415,258	
				Dark	120,874,869 ± 42,373,390	0.05
				Blue	12,535,775 ± 3,810,434	
33	707.9	L-Glutamine	C_5_H_10_N_2_O_3_	FL	294,061,961 ± 52,227,860	
				Dark	1,347,690,969 ± 20,051,935	0.000
				Blue	285,096,879 ± 30,214,602	
34	662.9	Asparagine	C_4_H_8_N_2_O_3_	FL	51,785,180 ± 5,177,064	
				Dark	150,042,799 ± 25,531,561	0.02
				Blue	105,437,258 ± 14,148,732	0.03
35	781.7	L-Tyrosine	C_9_H_11_NO_3_	FL	16,119,127 ± 1,204,190	
				Dark	135,002,280 ± 26,479,454	0.02
				Blue	28,514,941 ± 3,558,700	0.02
36	932.3	L-Tryptophan	C_11_H_12_N_2_O_2_	FL	8,223,776 ± 1,547,891	
				Dark	70,186,274 ± 6,300,278	0.003
				Blue	16,435,122 ± 782,277	0.02
Carbohydrates						
37	725.7	D-Mannose	C_6_H_12_O_6_	FL	107,738,818 ± 84,582,603	
				Dark	543,957,084 ± 46,193,939	0.02
				Blue	209,192,752 ± 48,618,900	
38	733.3	L-sorbopyranose	C_6_H_12_O_6_	FL	316,158,562 ± 20,999,619	
				Dark	697,213,182 ± 20,642,808	0.001
				Blue	383,562,829 ± 121,668,384	
39	755.2	D-Fructose	C_6_H_12_O_6_	FL	253,630,327 ± 18,247,145	
				Dark	107,158,254 ± 3,685,079	0.009
				Blue	214,588,479 ± 48,192,251	
40	759.7	D-(-)-Tagatose	C_6_H_12_O_6_	FL	235,944,910 ± 19,541,216	
				Dark	149,426,526 ± 7,315,272	0.03
				Blue	205,866,117 ± 43,378,085	
41	809.3	D-Erythrofuranose	C_4_H_8_O_4_	FL	ND	
				Dark	643,197,453 ± 38,582,722	0.001
				Blue	ND	
42	851	D-Galactose	C_6_H_12_O_6_	FL	6,796,773 ± 416,588	
				Dark	25,968,973 ± 686,2756	0.04
				Blue	6,104,887 ± 607,559	
43	876.4	Caffeic acid	C_9_H_8_O_4_	FL	ND	
				Dark	6,097,960 ± 405,390	0.001
				Blue	ND	
44	908.9	Methyl galactoside	C_7_H_14_O_6_	FL	ND	
				Dark	1,748,082 ± 248,700	0.008
				Blue	ND	
45	1020.5	Levoglucosan	C_6_H_10_O_5_	FL	3,822,565 ± 81,039	
				Dark	18,672,292 ± 536,7540	0.05
				Blue	7,096,685 ± 264,684	0.001
46	1160.6	Galactopyranose	C_6_H_12_O_6_	FL	1,243,798 ± 314,085	
				Dark	3,954,698 ± 428,917	0.01
				Blue	1,937,122 ± 233,853	0.02
Others						
d47	410	Dodecane	C_12_H_26_	FL	ND	
				Dark	2,089,678 ± 289,395	0.007
				Blue	750,133 ± 109,095	0.08
48	453.8	Ethanolamine	C_2_H_7_NO	FL	27,440,780 ± 624,097	
				Dark	1,463,234 ± 97,721	0.000
				Blue	22,216,551 ± 8,680,885	
49	533	Dopamine	C_8_H_11_NO_2_	FL	ND	
				Dark	1,463,234 ± 97,721	0.000
				Blue	2,322,461 ± 76,435	0.000
50	608	Penicillamine	C_5_H_11_NO_2_S	FL	ND	
				Dark	ND	
				Blue	2,036,085 ± 189,334	0.003
51	823.4	Palmitic Acid	C_16_H_32_O_2_	FL	12,165,495 ± 236,559	
				Dark	30,359,205 ± 4,795,821	0.02
				Blue	16,927,244 ± 2,657,439	
52	927	Myristic acid	C_14_H_28_O_2_	FL	7,061,525 ± 491,473	
				Dark	12,418,666 ± 1,559,438	0.02
				Blue	7,921,258 ± 489,889	
53	1467.7	5-Methyluridine	C_10_H_14_N_2_O_6_	FL	ND	
				Dark	8,401,912 ± 758,232	0.003
				Blue	3,838,136 ± 443,175	0.005

## Data Availability

The data presented in this study are available in Appendix A.

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
