# Peer review of "The Impact of Light Wavelength and Darkness on Metabolite Profiling of Korean Ginseng: Evaluating Its Anti-Cancer Potential against MCF-7 and BV-2 Cell Lines"

_ijms, 2023, doi:10.3390/ijms24097768_

Round 1

Reviewer 1 Report

The study is very good and original in terms of showing how secondary metabolites change with environmental conditions (especially light) and what consequences this causes. Congratulations

Author Response

Thank you for taking the time to review our manuscript. We greatly appreciate your positive feedback and are glad that you found our study to be both original and informative. We believe that understanding how secondary metabolites change in response to environmental conditions is crucial for unraveling the complex mechanisms underlying plant adaptation. We hope that our findings will contribute to a deeper understanding of plant physiology and help inform future research in this field. Thank you again for your encouraging comments.

Reviewer 2 Report

In this manuscript, the authors investigated the impact of darkness and different light wavelengths on the metabolomics and anti-cancer activity of ginseng extracts on in vitro cell model. Although the authors carried out extensive investigation, the claims of the manuscript are not convincing and not fully supported by the experiments conducted. The in vivo experiments for evaluating the therapeutic effects are essential for relevant research. Additionally, In the section "2.5. Dark-treated ginseng extract inhibits the pathway of NF-κB signaling", here a pathway map containing the proteins investigated is highly recommended to provide. It might be interesting research after an extensive revision and additional in vivo assays.

Author Response

Response to Reviewer 2 Comments

Point 1: The in vivo experiments for evaluating the therapeutic effects are essential for relevant research. Additionally, In the section "2.5. Dark-treated ginseng extract inhibits the pathway of NF-κB signaling", here a pathway map containing the proteins investigated is highly recommended to provide.

Response 1: Thank you for your feedback. We understand that in vivo experiments are crucial for evaluating the therapeutic effects of a treatment, however, due to various practical constraints, conducting such experiments may not always be feasible. In this particular study, we have made an effort to evaluate the effects of our treatment using in vitro assays and have provided relevant data to support our findings. We do agree that a pathway map containing the proteins investigated would be useful in providing a clearer picture of the mechanisms involved. We will consider adding this to our future work.

As this is our first report in this area, we have outlined the limitations of our study and have suggested further investigation in this area. We hope that our findings will pave the way for future research in this field and that our work will contribute to the development of novel therapeutic strategies. Thank you for your valuable feedback, and we will strive to improve our manuscript based on your comments.

Reviewer 3 Report

The use of ginsenosides as anticancer molecules is widely reported in the literature. This study investigates the antitumor potential of ginsenosides extracted from plants grown in hydroponics and subjected to different types of light: blue LED, dark light and white fluorescent (FL). The different extracts obtained after light stress were analyzed to determine the content of ginsenosides and other primary metabolites, and used to test their antitumor power on MCF-7 and BV-2 cell lines. Moreover, the mechanism of response of tumor cells (apoptosis) subjected to treatment with ginseng extracts was investigated.

The article is well written and interesting, the introduction is complete and clearly explains the background of the research topic; the data support the conclusions but could be better explained and represented.

I suggest the following revisions:

-        Lines 29-30: MCF-7 cells are not subjected to dark treatment therefore cell apoptosis is not induced by the treatment but by ginsenosides extracted from plant subjected to dark treatment. Rephrase the sentence

-        Line 111: figure 2 is mentioned in the text before fig 1. they must be reversed

-        Line 113: Table 1 reports all the area values of the identified compounds but most of them are already reported in figure 2. In my opinion, table 1 is redundant and too long (8 pages!) and does not provide any additional significant data. It should be placed as supplementary material.

-        Fig 2 and table 1 show the data expressed as area of the peaks but it is not correct! data should be expressed as concentration (% or mg/g dry material). In this way the response factor of the instrument is taken into account and also one or more standards (internal or external) must be used to perform the calibration.

-        Line 209 “PARP protein” what is this protein? The subchapter 2.4 explained fig.4 and must be united to the upper chapter too better understanding

-        Line 209:  “PARP protein” what is this protein?

-        Subchapter 2.4 explained fig.4 and should be merged with the previous chapter for a better understanding

-        Subchapter 2.5 and fig 5: - there are many terms that are not explained (NF-kB, ph-p65, LPS….)

-        Subchapters 2.5 and 2.6: the whole experiment summarized in fig 5 needs to be explained better in result section and in the material and method section.

-        Subchapter 4.5: How were the mass spectra of individual peaks identified? ( standards or the instrument library)

Author Response

Response to Reviewer 3 Comments

Point 1: Lines 29-30: MCF-7 cells are not subjected to dark treatment therefore cell apoptosis is not induced by the treatment but by ginsenosides extracted from plant subjected to dark treatment. Rephrase the sentence

Response 1: It's great to hear that you took the feedback and worked on it to address the issue raised.

Regarding the sentence in question, Lines 29-30 of abstract it has been repharesed to “The dark treated ginseng extract significantly induced apoptotic signaling in MCF-7 cells and dose-dependently inhibited the NF-κB and MAP kinase pathways in LPS-induced BV-2 cells” Since MCF-7 cells are not exposed to dark treatment, the induction of cell apoptosis is not due to the treatment itself, but rather to the ginsenosides derived from plants that have undergone such treatment. This rephrased sentence clarifies that the apoptosis is not caused by the dark treatment applied to the cells but rather by the bioactive compounds derived from the plant material that was subjected to dark treatment.

Point 2: Line 111: figure 2 is mentioned in the text before fig 1. they must be reversed

Response 2: Thank you for bringing this to my attention. If fixed it as directed. I removed figure 2 mentioned in Line 111, as it was explained later in the chapter.

Point 3: Line 113: Table 1 reports all the area values of the identified compounds but most of them are already reported in figure 2. In my opinion, table 1 is redundant and too long (8 pages!) and does not provide any additional significant data. It should be placed as supplementary material.

Response 3: While it is true that some of the information presented in Table 1 may already be included in Figure 2, it is important to note that the table provides more detailed information about the compounds in each treatment separately. This can be especially useful for researchers who may want to examine the data more closely or compare specific compounds across treatments. Additionally, the table allows for easier data extraction and analysis, which may not be as feasible when examining the figures alone. However, if you still feel that this table will suit more in supplimentary I will consider this change.

Point 4: Fig 2 and table 1 show the data expressed as area of the peaks but it is not correct! data should be expressed as concentration (% or mg/g dry material). In this way the response factor of the instrument is taken into account and also one or more standards (internal or external) must be used to perform the calibration.

Response 4: Thank you for your comment regarding the data presentation in Figure 2 and Table 1. You are correct that the peak areas alone do not provide an accurate representation of the actual amounts of the compounds analyzed. However, it is common practice in GC-MS analysis to express the data as relative peak areas, which are then used for comparison between samples. In this case, the internal standard, ribitol, was used to correct for variations in injection volume and other instrumental factors, which takes into account the response factor of the instrument. The use of relative peak areas in recent published articles for standardization of peaks when GC-MS is used is also a widely accepted practice.

Furthermore, the use of relative peak areas allows for the determination of the significance of the differences observed between treatments, as you have pointed out. While absolute quantification of the compounds would be ideal, it is often not feasible due to the complexity of the matrix and the need for authentic standards. Therefore, relative quantification using an internal standard is a reliable method for comparing the relative amounts of compounds between different samples or treatments.

In summary, the use of relative peak areas with an internal standard, such as ribitol, is an accepted method for the quantification of compounds in GC-MS analysis. The use of this method allows for the determination of the significance of differences observed between treatments and is widely used in the scientific community.

Following are few examples in which researchers used RPA as standardization.

  1. Xin, X., Pang, S., de Miguel Mercader, F., & Torr, K. M. (2019). The effect of biomass pretreatment on catalytic pyrolysis products of pine wood by Py-GC/MS and principal component analysis. Journal of Analytical and Applied Pyrolysis138, 145-153.
  2. Jonsson, P., Johansson, A. I., Gullberg, J., Trygg, J., Grung, B., Marklund, S., ... & Moritz, T. (2005). High-throughput data analysis for detecting and identifying differences between samples in GC/MS-based metabolomic analyses. Analytical chemistry77(17), 5635-5642.
  3. Baldermann, S., Yang, Z., Katsuno, T., Tu, V. A., Mase, N., Nakamura, Y., & Watanabe, N. (2014). Discrimination of green, oolong, and black teas by GC-MS analysis of characteristic volatile flavor compounds. American Journal of Analytical Chemistry2014.

Point 5: Line 209 “PARP protein” what is this protein? The subchapter 2.4 explained fig.4 and must be united to the upper chapter too better understanding

Point 6: Line 209:  “PARP protein” what is this protein?

Point 7: Subchapter 2.4 explained fig.4 and should be merged with the previous chapter for a better understanding

Response 5, 6, 7:

The subchapters 2.4 and 2.5 have been merged into 2.4 as per the directions of the reviewer. For a more detailed explanation of the PARP protein and its mechanism of action, a detailed role has been added to the manuscript. Please refer to lines 211-234 in the revised version of the manuscript.

PARP (Poly (ADP-ribose) polymerase) is a family of enzymes that are involved in various cellular processes such as DNA repair, genomic stability, transcriptional regulation, and programmed cell death. The PARP protein family includes 17 members, with PARP-1 being the most extensively studied member.

PARP-1 is involved in the repair of single-strand DNA breaks and plays an important role in maintaining genomic stability. PARP inhibitors are a class of drugs that are being developed as cancer therapies, as they have been found to be effective in killing cancer cells that have specific DNA repair defects.

In addition to its role in DNA repair, PARP-1 also regulates transcription by modifying chromatin structure and recruiting transcription factors to gene promoters. PARP-1 has also been implicated in the regulation of cell death, with studies suggesting that it can promote either apoptosis or necrosis depending on the cellular context.

Point 8: Subchapter 2.5 and fig 5: - there are many terms that are not explained (NF-kB, ph-p65, LPS….)

Point 9: Subchapters 2.5 and 2.6: the whole experiment summarized in fig 5 needs to be explained better in result section and in the material and method section.

Response 8, 9: We apologize for any confusion caused by the use of technical terms without proper explanation. We have updated the relevant sections of the manuscript to provide clear definitions of all technical terms used. kindly refer to M&M line 447- 459, 471-490. Necessary revisions has been done to clarify the methodology used and provide a more detailed explanation of the results.

Point 10: Subchapter 4.5: How were the mass spectra of individual peaks identified? ( standards or the instrument library)

Response 10: we used instruemntal library for peak area identification followed by MS data analysis of individual compounds. Rabitol was used as internal starndard for calibration. Latest version NIST 20 was used for peak detection.

We appreciate your feedback and thank you for helping us improve the clarity and accuracy of our manuscript. Please let us know if you have any further suggestions or concerns.